# The Central Nervous Mechanism of Stress-Promoting Cancer Progression

**DOI:** 10.3390/ijms232012653

**Published:** 2022-10-21

**Authors:** Yuchuan Hong, Lu Zhang, Nian Liu, Xirong Xu, Dan Liu, Jie Tu

**Affiliations:** 1Shenzhen Key Laboratory of Neuroimmunomodulation for Neurological Diseases, CAS Key Laboratory of Brain Connectome and Manipulation, The Brain Cognition and Brain Disease Institute (BCBDI), Shenzhen Institute of Advanced Technology (SIAT), Chinese Academy of Sciences (CAS), Shenzhen 518055, China; yc.hong@siat.ac.cn (Y.H.); lu.zhang1@siat.ac.cn (L.Z.); nian.liu@siat.ac.cn (N.L.); xirong.xu@siat.ac.cn (X.X.); liudan@siat.ac.cn (D.L.); 2Shenzhen-Hong Kong Institute of Brain Science-Shenzhen Fundamental Research Institutions, Shenzhen 518055, China; 3University of Chinese Academy of Sciences, Beijing 100049, China; 4Faculty of Life and Health Sciences, Shenzhen Institute of Advanced Technology (SIAT), Chinese Academy of Sciences (CAS), Shenzhen 518055, China; 5Guangdong Provincial Key Laboratory of Brain Connectome and Behavior, The Brain Cognition and Brain Disease Institute (BCBDI), Shenzhen Institute of Advanced Technology (SIAT), Chinese Academy of Sciences (CAS), Shenzhen 518055, China

**Keywords:** cancer, central nervous system, sympathetic nervous system, adrenergic receptor, neuropeptide Y receptor, HPA axis, glucocorticoid receptor, stress

## Abstract

Evidence shows that stress can promote the occurrence and development of tumors. In recent years, many studies have shown that stress-related hormones or peripheral neurotransmitters can promote the proliferation, survival, and angiogenesis of tumor cells and impair the body’s immune response, causing tumor cells to escape the “surveillance” of the immune system. However, the perception of stress occurs in the central nervous system (CNS) and the role of the central nervous system in tumor progression is still unclear, as are the underlying mechanisms. This review summarizes what is known of stress-related CNS-network activation during the stress response and the influence of the CNS on tumors and discusses available adjuvant treatment methods for cancer patients with negative emotional states, such as anxiety and depression.

## 1. Background

Stress is an exogenous or endogenous stimulation to individual emotion that can be found almost anywhere, and it is thus important that animals regulate their bodily functions in response to stressors [1]. However, once exposed to excessive stress, or long-term chronic stress, anxiety develops and damage to health is incurred [2]. Many brain regions are involved in the regulation of stress and anxiety, including the bed nucleus of the stria terminalis (BNST), central amygdala (CeA), basolateral amygdala (BLA), media prefrontal cortex (mPFC), nucleus of paraventricular hypothalamus (PVN), locus coeruleus (LC), and the periaqueductal grey (PAG) [3]. These regions are interconnected and form a complex network in which almost every region has the potential to influence others via mono- or polysynaptic connections. Therefore, when excessive chronic stress stimulates certain brain regions within this network, it may result in activation across the entire neural network, and long-term stress from either single or multiple factors may lead to mental disorders by altering neural network activity and signal transduction.

Anxiety disorder is a common mental illness with a high incidence and has a negative impact on human health and social stability. In China, both lifetime prevalence (7.6%) and 12-month prevalence (5.0%) of anxiety rank first amongst all mental illnesses [4]. Anxiety does not only affect mental health, but also physical health; the range of effects due to stress is very broad [5]. Numerous studies have shown that anxiety influences many functions, such as sensation, digestion, sexual function, immunity, and cancer progression [6,7,8,9,10]. There are a number of ways by which anxiety promotes cancer progression. A recent meta-analysis of cancer patients found that anxiety promotes tumor growth and metastasis, induces treatment resistance and relapse, and reduces survival [11]. 

How, then, do anxious states affect tumor growth? During states of anxiety, central brain regions involved in the stress response are activated, followed by activation of associated descending pathways, which results in abnormal secretion of various neurotransmitters, hormones, and other factors, thereby affecting tumor cells and their microenvironment [12]. Studies have shown that the sympathetic nervous system (SNS) and the hypothalamic–pituitary–adrenal (HPA) axis are the two main systems through which stress-related brain regions regulate peripheral circumstances. The sympathetic nervous system is part of the autonomic nervous system, which controls the body’s “fight or flight” response to external stimuli and can be activated quickly in response to stress. Activation of the sympathetic nervous system rapidly increases heart rate, respiratory rate, and blood pressure, thereby causing a state of arousal [13]. In contrast, the HPA axis is a part of the neuroendocrine system. It is activated minutes or hours after a stressor to regulate metabolism and provide energy [14]. In cancer patients with anxiety disorder, both the sympathetic nervous system and the HPA axis are abnormally activated, and neurotransmitters and hormones are secreted into the vicinity of the tumor or in the systemic circulation. These molecules promote tumor progression by regulating cell proliferation, survival, angiogenesis, and immune responses [15,16,17]. 

Most current research focuses on the effects of the secretion and action of specific molecules and hormones on cancer progression after activation of the sympathetic nervous system and/or the HPA axis [15,18,19,20]. However, there is a lack of research on the central neural mechanisms underlying tumorigenesis and tumor development caused by long-term stress. Stress perception occurs in the central nervous system and investigation of the central neural mechanisms underlying stress-induced tumor progression can provide new insights into putative cancer therapeutics. In this paper, we review and summarize the interaction of stress-activated central neural circuits and associated descending pathways with peripheral tumor and tumor microenvironments. We also discuss the effects of various neurotransmitters, hormones, and other factors abnormally secreted under anxious states, on cancer progression. Finally, limitations of the current research and the prospects of future clinical applications of this fundamental research are discussed.

## 2. Stress Accelerates Tumor Progression via the Sympathetic Nervous System

The sympathetic nervous system originates in the ventral brainstem, where sympathetic premotor neurons are found. They are found predominantly in the rostral ventrolateral medulla (RVLM) and in the rostral ventromedial medulla (RVMM). These neurons project to the intermediolateral nucleus (IML, also known as the sympathetic preganglionic nucleus), which then projects to the dorsal root ganglia (DRG) for terminal output to peripheral organs which control heart rate, blood pressure, respiration, glycemia, vigilance and other physiological responses [21]. When negative emotions are induced under chronic stress, the sympathetic nervous system is continuously activated and increases the release of catecholamines (such as epinephrine and norepinephrine) [22,23]. In a spontaneous colon tumor model, ablation of sympathetic premotor neurons in *APC^min/+^* mice reduces the number of polyps in the mouse intestine [24]. Sympathetic denervation also leads to decreased tumorigenesis in a spontaneous prostate tumor mouse model [25]. These results suggest that loss of SNS function may slow tumorigenesis (Figure 1).

### 2.1. Stress Activates SNS-Related Neural Circuits

The RVLM is associated with tumor growth. Recently, Zhang and colleagues found that pharmacogenetic tools, DREADDs (Designer Receptors Exclusively Activated by Designer Drugs), used to manipulate catecholamine neurons in the RVLM, regulate CD8^+^ immune cells and promote immune evasion [24]. Under anxious states, neurons in the RVLM undergo a similar activation by receiving signals from upstream anxiety-regulation brain regions, indicating that anxiety-promoted tumor progression may be achieved via activation or inhibition of neural circuits projecting to the RVLM from anxiety-associated brain regions [26]. In addition, the RVMM also controls temperature and pain. It receives complex inputs from the whole brain, including antinociception information from the PAG and thermogenic information from the dorsal media hypothalamus (DMH). However, it is not clear whether the RVMM or RVMM-associated circuits are involved in the regulation of cancer progression during the stress response.

Anxiety is a state of arousal that occurs in response to stress. The amygdala, including the BLA, the CeA, the medial amygdala (MeA) and the BNST (extend amygdala), is considered to be an important brain area for processing stress [3]. Experimental activation of the amygdala and its downstream projection targets, including the lateral hypothalamus (LHA), the LC, the PAG and other regions, results in an anxious state [27]. Tumor studies have suggested the association between the activity of the amygdala and cancer: a study of cancer patients found that the left amygdala volume is larger in patients with a psychiatric history compared to those with no such history [28]. Investigations using fMRI have shown that amygdala activity in breast cancer patients is associated with peripheral inflammatory factors and that social support reduces amygdala activity and lowers levels of inflammatory markers [29,30]. Therefore, these studies may indicate the association between cancer and amygdala activity. To explain these findings, it is thought that amygdala activation is highly involved in sympathetic activity, and that neurons in the amygdala that project to areas containing sympathetic premotor neurons have anatomical and functional overlap with those regions which elicit anxiety responses. It is known that activation of somatostatin^+^ GABAergic neurons in the CeA regulates blood pressure and other sympathetic functions by projecting to the RVLM (sympathetic premotor area) or the nucleus of the solitary track (NTS, peripheral sensory center) [31]. This means that activation of the CeA directly leads to sympathetic activation. The BNST is also involved in stress-induced anxiety [32], and there are direct or indirect projections from the BNST to the medulla, which regulate sympathetic function [33,34,35,36]. 

The hypothalamus, including the DMH and the LHA, is a downstream output target of the amygdala and cortex, which is also involved in encoding anxiety information. It is thought to play an important role in regulating sympathetic activity during stress. This area is a crucial hub for projections to regions containing sympathetic premotor neurons. During stress, the amygdala inhibits the ventral DMH, disinhibits the GABAergic projection from the ventral DMH to the medulla, where the sympathetic premotor neurons are activated, resulting in sympathetic functions [37]. Orexin/hypocretin neurons within the LHA are thought to be involved in stress. Activation of the orexin system induces anxiety-like behavior [38]. In addition, the orexin system is found to be associated with breast cancer in animal models: activation of LHA orexin neurons in a mouse model of breast cancer leads to sleeping disruption and metabolic abnormality complicated by tumors, and this effect occurs via the sympathetic system as it can be blocked by 6-hydroxydopamine, a selective catecholaminergic neurotoxin [39].

An earlier comparative study of animal models showed that periaqueductal gray (PAG) activity is associated with breast tumor growth [40]. Indeed, the PAG is closely related to the regulation of cancer pain [41,42]. The important descending pain pathway, the PAG-RVMM projection, which extends to the dorsal horn, is the primary pathway for pain suppression [43]. Regulation of nociception is influenced by anxiety circuits which are modulated by the amygdala. GABAergic neurons in the amygdala project to PAG GABAergic neurons and locally innervate adjacent glutamatergic neurons. Following chronic inhibitory stress, inhibitory signaling by these amygdala projections relieves GABAergic inhibition of glutamatergic neurons in the PAG, thereby regulating nociception [44]. In addition, the PAG is also involved in sympathetic functions: activation of the lateral/dorsolateral PAG is known to increase heart rate and arterial pressure [45]. Therefore, the PAG may be involved in the regulation of tumor progression through pain regulation pathways and sympathetic pathways.

The mPFC is at the top of the response initiation hierarchy during the stress response. It has functional links that govern the amygdala and hippocampus [37]. The mPFC is considered the region that suppresses anxiety. For instance, activation of glutamatergic projections from the mPFC to the amygdala causes anxiolytic effects, whereas inhibition results in anxiogenic effects [46]. At the same time, activation of these regions inhibits stress-induced sympathetic activity [47]. However, no direct connection between sympathetic premotor neurons and the mPFC has been found. Reward signals may lead to mPFC activation [48], and activation of reward circuits involving the ventral tegmental area (VTA) is thought to reduce negative emotion [49]. Studies investigating tumors have found that activation of the VTA also promotes immune function, resulting in inhibition of tumor growth in mice [50]. Our previous study found that activation of the dopaminergic projections from the VTA to the mPFC reduces anxiety levels in stressed animals, and tumor growth slows down as anxiety levels decrease [51]. At the same time, anxiety-related sympathetic hormone levels also decrease, indicating the importance of the mPFC in tumor regulation and treatment. 

### 2.2. Sympathetic Nerve Fibers Release Neurotransmitters to Promote Tumor Progression

Sympathetic nerve fibers originate from the DRG and project to nearly all organs and tissues, including solid tumors. In addition to the original neuronal fibers in pathological tissues, newly formed neuronal fibers also develop during the early cancer states [25,52]. Long-term, continuous, specific activation of sympathetic nerve fibers of the mice around tumors using NaChBac-channel viruses significantly increases catecholaminergic neurotransmitter levels and promotes cancer growth and metastasis, with adrenalectomy, indicating an important role of sympathetic nerve fibers in cancer progression [53]. The major secretions of sympathetic nerves are norepinephrine (NE) and neuropeptide Y (NPY).

The focus of recent research into the regulation of stress-related cancer progression has been NE and NE signaling since Thaker and colleagues found that NE and β-adrenergic receptor (β-AR) signaling induced by elevated chronic stress promotes tumor growth and angiogenesis in mice [54]. NE release during the stress response is thought to contribute to increased DNA damage and cause tumorigenesis [55,56]. NE activates arrestin-β and the PKA system, further resulting in p53 inactivation and inhibits p53-mediated DNA damage repair [57]. Spontaneous tumor model studies have also demonstrated the negative effects of stress-induced DNA damage on tumor therapy [58]. The adrenergic receptor antagonist ICI 118,551 and β-AR knockout blocks cancer development caused by chronic restraint, thereby reducing the proportion of pancreatic ductal adenocarcinoma (PDAC) in LSL-*Kras*^+/G12D^; *Pdx1*-Cre (KC) mice, a spontaneous pancreatic tumor model, while the agonist isoproterenol promotes PDAC [59]. NE-β-AR signaling activates many biological reactions and cell-signaling-related proteins, such as Src and CREB, and also activates L-type voltage-dependent calcium channels (VDCC) [59,60,61]. These reactions promote cancer proliferation. NE-β-AR signaling is also necessary for angiogenesis as it results in an energy acquisition switch from oxidative phosphorylation to glycolysis in endothelial cells, and thus angiogenesis [62]. The immune response is closely related to the development and treatment of cancer. NE-β-AR signaling stimulates macrophage development, differentiation, polarization to M2, infiltration, and therefore promotes cancer metastasis [17,63,64,65]. 

NPY is another neurotransmitter released by sympathetic nerve fibers in response to stress. Levels of NPY remain elevated longer than NE does during stress responses and sympathetic activation [66]. However, in contrast to NE-receptors signaling, NPY has not been adequately investigated in tumor studies. In vitro studies have shown that NPY can activate Y5R or the Y2R–Y5R complex to promote cell proliferation via the Erk pathway [67,68]. In addition, Y2R activated by NPY in endothelial cells promotes angiogenesis [69,70]. Macrophages, which express large amounts of Y1R, are also affected by NPY. Activation of Y1R in macrophages leads to the release of NO and cytokines, including IL-4, IL-6, IL-12, and TNF-α, which promote inflammation and angiogenesis [71] (Figure 2).

### 2.3. The Adrenal Medulla Secretes Epinephrine to Promote Tumor Progression

The adrenal glands are activated in response to stress. They are controlled by sympathetic projections to the adrenal medulla. In response to stress, two hormones, epinephrine and NE, are released and enter the circulation [72].

The adrenal medulla predominantly releases epinephrine (~75%) [73]. During acute stress, epinephrine is released in large quantities, improving the ability to deal with danger. Epinephrine and NE share receptors, so epinephrine also has a negative impact on cancer development. Epinephrine leads to cell proliferation by adrenergic receptors [74] and binding to β-ARs activates the PKA system and further regulates BAD and MCL-1 proteins to inhibit apoptosis [75,76]. Epinephrine-β-AR signaling also promotes cancer stem-like traits through a cascade of responses produced by lactate, which is metabolized by LHDA [77]. Epinephrine also promotes angiogenesis: activation of epinephrine-β-ARs-HIF-1α results in increased VEGF secretion [78]. An immunological study found that the elevation of epinephrine caused by social disruption suppresses CD8^+^ T-cell proliferation as well as macrophage-derived IFN-γ [79] (Figure 2).

## 3. Stress Accelerates Tumor Progression via the HPA Axis

The hypothalamus–pituitary–adrenal (HPA) axis is a classical hormone regulation pathway. It is an important part of the neuroendocrine system, involved in controlling responses to stress and regulating many different physical activities. When individuals suffer from stress, projections from the hypothalamus to the median eminence (ME) activate the release of the adrenocorticotropic hormone (ACTH) from the pituitary gland and ultimately results in the release of glucocorticoids (GCs) from the adrenal cortex into the circulation [80]. Stress promotes activation of the HPA axis, and hyperactivity within the HPA axis is related to the poor prognosis of cancer patients [81]. The role of the HPA axis in the regulation of cancer is predominantly through the release of GCs (Figure 3).

### 3.1. Stress Activates HPA Axis-Related Neural Circuits

The PVN is a subregion of the hypothalamus and is considered to be the origin of the HPA axis. Corticotropin-releasing hormone (CRH) neurons located in the PVN become activated under stress and secrete CRH through the ME to the third ventricle, which stimulates the pituitary, resulting in the secretion of ACTH [80]. The PVN itself is thought to be associated with the stress-related response. Specific knockout of PVN^CRH^ neurons in mice results in anxiolytic behaviors [82]. Interestingly, sucrose used as a reward inhibits the activity of CRH neurons and reduces anxiety [83]. Projections to the PVN from various brain regions, including the amygdala and the mPFC, the brainstem, and other hypothalamic brain regions, also affect PVN activation. These projections are activated under stress, thereby promoting the release of CRH, and further, ACTH and GCs, from the HPA axis, which ultimately act on peripheral organs [84]. The PVN, in addition to the neuroendocrine system, is also involved in the sympathetic network. These PVN neurons that project to the RVLM and the IML directly control glucose metabolism, blood pressure, and other physiological processes [26,85]. 

The PVN plays an important role in the regulation of internal stress. This stress is largely derived from changes in the peripheral environment. Afferent autonomic nerves sense the peripheral environment through the sympathetic and parasympathetic systems, similar to the efferent autonomic nerves, and this input is then sent to the brain. The NTS, which is in the brain stem, integrates this information from the sympathetic and parasympathetic systems and projects to the PVN. These signals include abnormalities from various visceral lesions and fluid imbalances. These abnormalities can cause a stress response similar to external psychological input [86]. Excitatory glutamatergic neurons in the NTS project to the PVN and their excitation activates the HPA axis, resulting in the secretion of ACTH and GCs [87]. The PVN also receives projections from the subfornical organ (SFO), a region involved in the regulation of fluid balance and blood pleasure. The SFO contains angiotensin II projections to CRH neurons of the PVN and activates the HPA axis via angiotensin II receptors [88,89]. This suggests that peripheral stress signals activate PVN^CRH^ neurons and the HPA axis via sensory input regions.

The hypothalamus also regulates the PVN during the stress response. There are glutamatergic neurons in the posterior hypothalamus (PH) that project to the PVN [90]. Functional studies have shown that inhibition of the PH via injection of a GABAa agonist significantly reduces ACTH release [91] and corticosterone responses to acute depression and auditory stress [92]. These studies demonstrate the role of hypothalamic activation in the stress-induced excitability of the HPA axis. Other investigations of the DMH have shown that it innervates PVN^CRH^ neurons within the HPA axis. Interneurons in the DMH receive projections from the CeA and MeA regions of the amygdala. These inhibitory projections predominantly originate in the ventral DMH [93], which in turn sends GABAergic projections which inhibit the PVN [94]. Activation of the amygdala during the stress response relieves the inhibitory effect of the ventral region on the PVN and HPA axis, thereby promoting ACTH release [37]. 

Most amygdala innervations to the hypothalamic axis are from the MeA and the CeA. These subregions lack substantial direct connection with the PVN, so amygdala regulation of the PVN and the HPA axis is predominantly through interneuron disinhibition. In addition to the DMH projections mentioned above, the MeA sends a GABAergic projection to the peri-PVN, an area surrounding the PVN which has GABAergic neurons [95]. The MeA activates the PVN and HPA axis by de-inhibiting these neurons. 

The mPFC acts as a suppressor in its regulation of the HPA axis. Lesions of the infralimbic (IL) and prelimbic (PL) cortical subregions of the mPFC promote the secretion of ACTH and GCs [96]. In addition, inhibition of neuronal activation in the IL using siRNA increases ACTH release during the stress response [97]. Another study showed that excitatory neurons in the PL attenuate the HPA axis via anteroventral BNST GABAergic neurons, which inhibit the PVN and the HPA axis [98]. The mPFC is also an important region for negative feedback regulation of the HPA axis. The mPFC is abundant in glucocorticoid receptors [99]. These receptors excite glutamatergic mPFC neurons which attenuate the PVN and HPA-axis activation. Knockdown of this receptor increases HPA responsiveness to stress [100]. Activation of the VTA-mPFC pathway in mice leads to a reduction in the level of circulatory GCs, indicating an inhibitory effect of this circuit on the HPA axis resulting in slower cancer growth [51].

### 3.2. The Adrenal Medulla Secretes Glucocorticoids to Promote Tumor Progression

Glucocorticoids release from the adrenal cortex and perform their roles via the nuclear glucocorticoid receptor (GR). After binding to GCs, the GR homodimerizes, and the dimer translocates to the nucleus. In spontaneous tumorigenesis models, social isolation elevates GC levels and increases the size, number, distribution, and malignancy of spontaneous mammary tumors [101]. 

Stress-induced GCs promote ionizing radiation-induced tumorigenesis by reducing tumor suppressor p53 protein levels and down-regulating the tumor suppressor gene BRCA1 [102,103]. GC–GR signals also promote cancer progression via the Hippo pathway by regulating YAP and TEAD4 [104,105]. GCs also promote metastasis by acting on distant metastatic sites [16]. In immune reactions, GCs have a significant inflammation suppression effect when used as immunosuppressant. GCs were found to inhibit NK cells and stimulate CD8^+^ T-cell differentiation, causing immune dysfunction which defeats the immunity-checkpoint response and promotes tumor growth [106,107]. Therefore, GCs impair the efficacy of chemotherapy and immunotherapy [108]. 

The neurotransmitters, hormones and related signaling pathways involved in stress-induced tumor progression are summarized in Figure 2.

## 4. Other Stressors

Aging has been acknowledged as a major risk factor for developing cancer. Telomere was shortened with aging and each cell division, which is a hallmark of cellular senescence [109]. Short telomeres are associated with genomic instability, which is the main cause of tumorigenesis [110]. Alongside cellular senescence, aging also impairs immune functions, called immunosenescence, and leads to invalidation of the immune system against cancer [111]. The expression of CD27 and CD28, the markers of T-cell activation, is lower in CD57^+^ (senescence marker) CD4^+^, and CD8^+^ T cells [112,113]. However, M2 macrophages, which are thought to promote cancer progression via infiltration and angiogenesis, are increased in the old individuals [114,115]. PD-L1 can drive immune cell inactivation as the ligand of PD-1. A study has found that PD-L1 is upregulated in the senescent cells. This may help to explain the increased cancer incidences in the elderly population [116].

Oxidative stress is thought as a cause of aging [117]. It is caused by an imbalance between the production of reactive oxygen species (ROS) and the antioxidant capability [118]. Cancer cells have an inherently elevated ROS level compared to their normal counterparts [119]. It is noted that oxidative stress can promote cancer progression in tumorigenesis, proliferation, angiogenesis, and metastasis [120]. ROS has been found to activate Ras oncogene and inhibit p53, the tumor suppressor, to induce tumorigenesis [121,122]. Other tumor suppressor genes, such as cyclin-dependent kinase inhibitor 2A (CDKN2A), retinoblastoma (Rb), Von Hippel–Lindau (VHL), and breast cancer 1 (BRCA1) have also been identified in the cancer cells as being inactivated via an ROS-dependent epigenetic modulation [123,124]. In proliferation studies, the p66Shc protein level, ErbB-2 level, and Erk/MAPK activation has been elevated by increased ROS in cancer for cell proliferation [125]. PI3K/Akt pathway plays various roles in ROS-promoting cancer progression. At first, it is another cell pathway activated by ROS. It has been reported to inactivate PI3K/Akt phosphatases, such as phosphatase and tensin homolog (PTEN) and protein tyrosine phosphatase 1B (PTP1B) which may promote proliferation [126]. Oxidative stress is also necessary for angiogenesis. Activation of PI3K/Akt and MAPK pathway by ROS does not only promote proliferation but also induces the release of VEGF [127]. Transcription factor HIF-1α is another reason that promotes VEGF expression: ROS increases the HIF-1α expression and also inhibits prolyl hydroxylase and leads to the stabilization of HIF-1α [128]. The other function of ROS-activating PI3K/Akt and MAPK/Erk is inducing metastasis, i.e., ROS mediates HGF-driven invasion of cancer cells via Erk 1/2 activation. A previous study has shown that HGF regulates ROS-induced expression of urokinase plasminogen activator (uPA), a serine protease involved in cellular invasion, via the Erk 1/2 pathway, and it stimulates the invasiveness of human gastric cancer cells [129]. Oxidative stress and ROS also impair the immune system. Inhibition of oxidative metabolism, and production of ROS, can block the process of tumor cell induced-myeloid-derived suppressor cells (MDSCs) on the growth of colon cancer cells [130]. This may be due to the inhibition of ROS suppressing the negative effect of MDSCs in T cells and rescuing the activity of T cells [131]. On the contrary, high levels of ROS inhibit T cell activity by suppressing the formation of T cell receptor and major histocompatibility complex antigen complex [132]. ROS may also be involved in PD-L1 therapy. Chemotherapy drugs, paclitaxel or antioxidant depletion, upregulates ROS production and further induces PD-L1 expression in the macrophages. PD-L1 positive macrophages have immune-suppressive interference with the efficacy of paclitaxel in vivo. Thereby ROS inhibitors may be adjunct to PD-L1 therapy [133], but further clinical data are needed to demonstrate it.

## 5. Perspective

In a recent meta-analysis of cancer patients, anxiety was found to have a negative effect on prognosis and treatment [11]. Cancer patients were more likely to have comorbid anxiety than healthy people [134]. However, research into the treatment of clinical anxiety in cancer patients has not received much attention. Currently, clinical treatment is predominantly aimed at β-ARs using β-blockers, addressing anxiety-related mechanisms [135]. β-blockers have been used in the treatment of many types of cancer. However, the results of various meta-analyses on β-blocker efficacy do not confirm that β-blockers have a significant effect on reducing cancer progression [136,137,138,139]. Stress-induced hormones are complex, and a single blocker may not completely eliminate the adverse effects of stress on cancer progression and treatment. 

The central nervous system controls various secretory systems. Manipulation of the central nervous system can modulate the secretion of cancer-promoting molecules and progression can be greatly reduced. In addition, hormone secretion is limited to normal levels following the elimination of stress, thereby reducing side effects. As our group found previously [51], modulation of the reward-related VTA-mPFC circuit effectively reduces anxiety levels in mice and simultaneously suppresses circulating NE and GCs, thereby slowing down cancer growth. However, this is technically difficult at present since the regulation of neural circuits is still at the stage of laboratory experiments. In this regard, more efforts are required to promote the translation of effective treatments in animal models, such as optogenetic manipulation of neural circuits, to the clinic.

The central nervous system is not only the center of regulation but also the center of sensation. It is sensitive to external stimuli and feeds stimuli back through neural networks. Stimulating animals with environmental factors can reduce anxiety. For example, environment enrichment (EE) in mice which includes a large space for activity and provides a ‘sports and entertainment facility’ leads to happier, less anxious mice [140]. A recent study showed that EE modulates β-ARs-induced immune responses, slows down tumor growth and improves immunotherapy efficacy [141]. Another study found that exposing mice to an ambient temperature of 30 °C lowers rates of tumor growth compared to mice at 22 °C. This can also be achieved by reducing the NE-β-AR signaling [142]. In addition, light treatment can be useful in psychiatric treatments of mood disorders and in pain management due to antinociceptive effects. Specific intensities of bright light passed normally through the retina can activate several central brain regions, such as the habenula, which is associated with depression, and the PAG, which is associated with pain, and projections from the ventral lateral geniculate nucleus and intergeniculate leaflet (vLGN/IGL) can inhibit the PAG, thereby regulating mood and analgesia [143,144]. In the future, it may also be used in adjuvant cancer therapy.

Based on the literature covered in this review, we argue that the neural circuits related to stress should be further dissected experimentally to uncover relevant mechanistic details which can ultimately be utilized to generate adjuvant therapies to improve the survival of cancer patients.

## Figures and Tables

**Figure 1 ijms-23-12653-f001:**
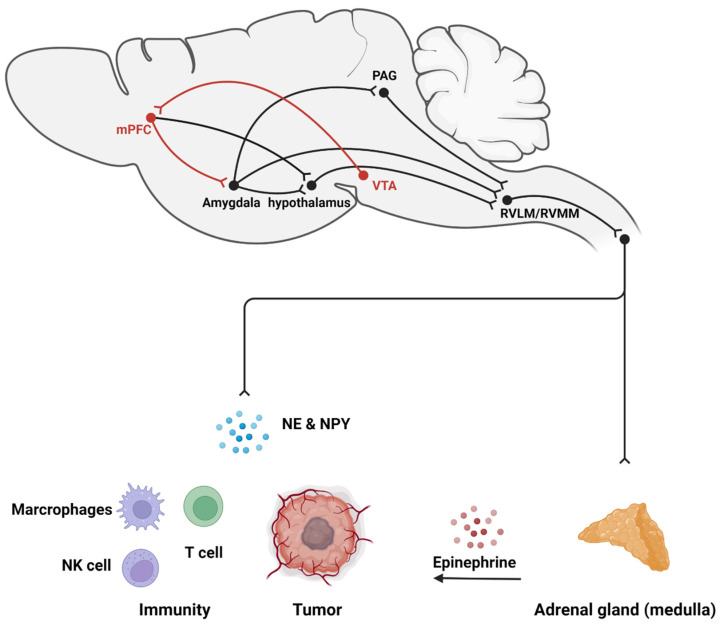
Stress promotes cancer progression via the sympathetic nervous system. During the stress response, anxiety-related circuits (black) are activated, which then activate the peripheral sympathetic nervous system through the sympathetic premotor regions RVLM and RVMM, leading to the release of NE and NPY into the tumor and its microenvironment. This promotes proliferation and angiogenesis, while causing immunosuppression. In addition, sympathetic excitation leads to the secretion of epinephrine from the adrenal medulla, which reaches the tumor through the circulatory system and promotes cancer progression. Activation of the reward system (red) can inhibit the effects of stress.

**Figure 2 ijms-23-12653-f002:**
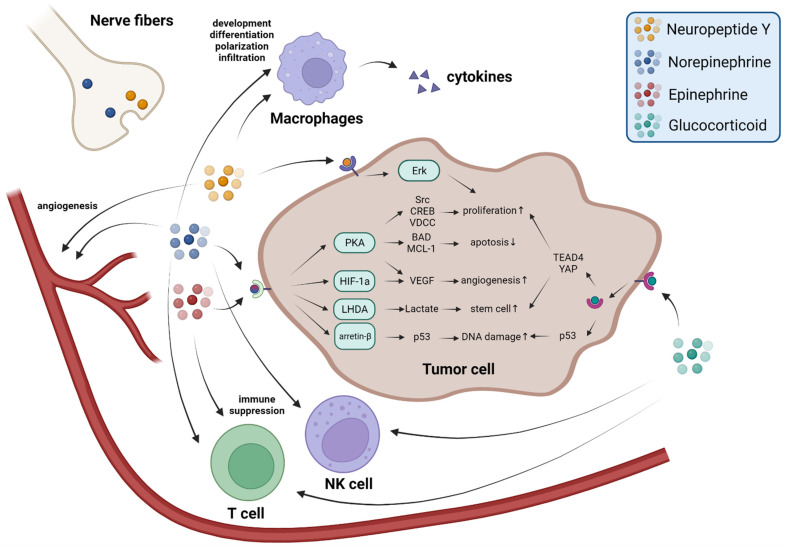
Stress-related neurotransmitters and hormones promote cancer progression in multiple ways. During the stress response, NE and NPY are released from nerve fibers, epinephrine is released from the adrenal gland medulla, GCs are released from the adrenal gland cortex and arrive at the tumor through the circulation. Their function occurs via receptors on cancer cells, blood vessels, and immune cells to promote cancer progression in multiple ways.

**Figure 3 ijms-23-12653-f003:**
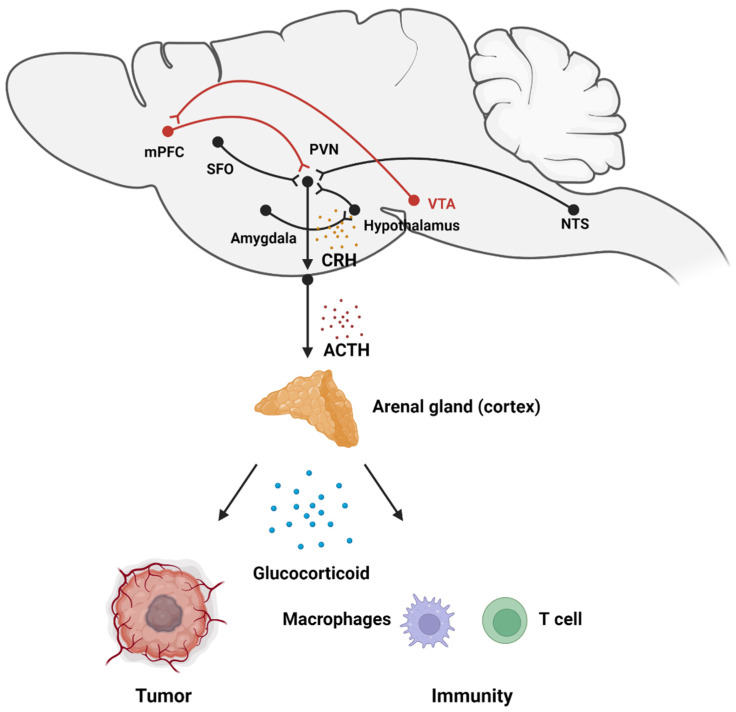
Stress promotes cancer progression via the HPA axis. During the stress response, anxiety-related circuits (black) are activated, which then activate the PVN, the origin of the HPA axis, resulting in the release of CRH and ACTH, and ultimately the release of GCs. GCs enter the circulation and act on the tumor and microenvironment, leading to immunosuppression, and ultimately promoting cancer progression. Activation of the reward system (red) can inhibit the effects of stress.

## Data Availability

Not applicable.

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
