# Peer review of "The Central Nervous Mechanism of Stress-Promoting Cancer Progression"

_ijms, 2022, doi:10.3390/ijms232012653_

Round 1

Reviewer 1 Report

In this review, the authors focus on the effects of various neurotransmitters, hormones, and other factors abnormally secreted under anxious states, on cancer progression. Finally, limitations of the current research and the prospects of future clinical applications of this fundamental research are discussed. The collect is very interesting, however, some minor problems, as indicated below, should be addressed before the document can be considered for publication in the this journal. This version of the manuscript is not enough complete.

Minor problems:

English language and style are fine, minor spell check is required to ensure that an international audience can clearly understand your text. In general, I suggest to modify the full text according to line guide of the journal. 

I suggest to introduce a new section in which the author's can discuss about the effects of oxidative stress. For example, oxidative stress increase promotes cancer progression, thus inhibiting the immune response. 

Another stressor would be natural aging. The authors should discuss also this phenomenon.

Author Response

Thank you for your kind comments. Please see the attachment.

Reviewer 2 Report

Dear authors,

You have done a careful and extensive review, but to my point of view there several points that need corrections or clarifications.

1. line 16 (abstract): “Stress can promote the occurrence and development of tumors.” This statement is very categorical. If think that, terms like, “Evidence shows that…” or “…could induce…”

2. line 26: stress should be defined

3. line 47: function instead of dysfunction    

4. line 57: As the authors explained later the HPA is not a descending pathway, but a part of the neuroendocrine system

5. line 106: Which pharmacogenetic tools?

6. lines 106 – 107: … “regulate CD8+ immune cells and promote cancer development.” I think that the term “immune evasion” fits better than development.

7. lines 120 – 130: The relation between the amygdala, inflammation and cancer should be better addressed, because the studies to which the authors referred are related with breast cancers survivors. The inflammatory state in such cases could also be caused by chemo- or radiation therapy. As also the issue of the association of the amygdala’s volume and psychiatric illness.

8. lines 144 – 176: Caution should be taken in this chapter, because the studies presented were animal studies and human tumours are different. This should also be explained.

9. lines 203 – 205: If macrophage infiltration produces tumour growth, these macrophages should be M2 (anti-inflammatory phenotype). This should be cleared.

10. line 228: “Epinephrine leads to cell proliferation.” It should be explained how epinephrine induces cell growth. Do tumour cells express adrenergic receptors? All tumour cells? Or does epinephrine induce tumour growth indirectly through mediators released by microenvironmental cells?

11. lines 243 -244: “Stress promotes activation of the HPA axis, and hyperactivity within the 243 HPA axis is related to the poor prognosis of cancer patients.” This statement needs documentation.

12. lines 263 -264: …” thereby promoting the release of CRH, and further, ACTH and GCs, from the HPA 263 axis, which ultimately act on peripheral organs.” Cancer patients receive frequently steroids, in consequence the HPA would be supressed and the levels of endogenous steroids would be low. This should also be taken in account.

13. line 266: blood pressure instead of hypertension.

14. lines 209 – 311: “Activation of the VTA-mPFC pathway leads to a reduction in the level of circulatory GCs, indicating an inhibitory effect of this circuit on the HPA axis resulting in slower cancer growth.” How can a reduction of glucocorticoids reduce cancer growth? Through a better immune response? This would also help to explain the statement of the following lines.

15. lines 315 -317: “In spontaneous tumorigenesis models, social isolation elevates GC levels, and increases the size, number, distribution, and malignancy of spontaneous mammary tumors”.

16. lines 325 -326: “Therefore, GCs increase the efficacy of chemotherapy and immunotherapy”. This statement is in contradiction with the action of steroids in reducing the immune response.

Author Response

(The authors gave the same response as above.)
